# High-Yield Expression and Purification of Recombinant Influenza Virus Proteins from Stably-Transfected Mammalian Cell Lines

**DOI:** 10.3390/vaccines8030462

**Published:** 2020-08-21

**Authors:** Jeffrey W. Ecker, Greg A. Kirchenbaum, Spencer R. Pierce, Amanda L. Skarlupka, Rodrigo B. Abreu, R. Ethan Cooper, Dawn Taylor-Mulneix, Ted M. Ross, Giuseppe A. Sautto

**Affiliations:** 1Center for Vaccines and Immunology, University of Georgia, Athens, GA 30602, USA; jwecker@uga.edu (J.W.E.); greg.kirchenbaum@immunospot.com (G.A.K.); spncr01@uga.edu (S.R.P.); skarlupka@uga.edu (A.L.S.); rbabreu@uga.edu (R.B.A.); recooper@uga.edu (R.E.C.); dawn.taylor-mulneix@nih.gov (D.T.-M.); tedross@uga.edu (T.M.R.); 2Department of Infectious Diseases, University of Georgia, Athens, GA 30602, USA

**Keywords:** recombinant protein, stable cell lines, plasmid, cloning, protein purification, IMAC, influenza, vaccine

## Abstract

Influenza viruses infect millions of people each year, resulting in significant morbidity and mortality in the human population. Therefore, generation of a universal influenza virus vaccine is an urgent need and would greatly benefit public health. Recombinant protein technology is an established vaccine platform and has resulted in several commercially available vaccines. Herein, we describe the approach for developing stable transfected human cell lines for the expression of recombinant influenza virus hemagglutinin (HA) and recombinant influenza virus neuraminidase (NA) proteins for the purpose of in vitro and in vivo vaccine development. HA and NA are the main surface glycoproteins on influenza virions and the major antibody targets. The benefits for using recombinant proteins for in vitro and in vivo assays include the ease of use, high level of purity and the ability to scale-up production. This work provides guidelines on how to produce and purify recombinant proteins produced in mammalian cell lines through either transient transfection or generation of stable cell lines from plasmid creation through the isolation step via Immobilized Metal Affinity Chromatography (IMAC). Collectively, the establishment of this pipeline has facilitated large-scale production of recombinant HA and NA proteins to high purity and with consistent yields, including glycosylation patterns that are very similar to proteins produced in a human host.

## 1. Introduction

Recombinant proteins are an accepted vaccine platform and are components in a range of commercially available vaccines including the hepatitis B (HBV) and human papilloma virus (HPV) vaccines [1,2,3]. Herein, we describe an approach for developing stable transfected human cell lines expressing recombinant influenza virus hemagglutinin (HA) and influenza virus neuraminidase (NA) proteins for the purpose of in vitro and in vivo vaccine development. HA and NA are the major glycoproteins expressed on the surface of influenza viral particles and the major targets of the humoral response [4,5]. The benefits of using recombinant proteins for in vitro and in vivo assays include ease of use, high level of purity and the ability to scale-up production. Additionally, vaccine strategies based on administration of recombinant proteins enable focusing of the immune response toward antigens associated with protective immunity [6]. Moreover, usage of recombinant proteins as immunogens circumvents issues associated with alternative influenza vaccine production systems, such as the occurrence of adaptive mutations using embryonated hen eggs for virus growth [7]. For this reason, current under-development next generation influenza vaccines mostly rely on a recombinant protein-based platform [7,8]. Additionally, the here described protein production platform can be considered safer and more affordable as compared, for example, to transducing systems such as those relying on recombinant lentivirus [9]. In fact, lentiviral-based systems often require a higher biosafety capacity [10].

Several vectors are commercially available for the expression of proteins in mammalian cell lines. As described in Figure 1, commercially available mammalian expression vectors to recombinantly express HA or NA (Figure 1) were utilized. Key features of these plasmids include a human cytomegalovirus (CMV) promoter for expression in mammalian cell lines and a polyadenylation signal and transcription termination sequence downstream of the multiple cloning site for enhanced mRNA stability. Furthermore, these plasmids have appropriate antibiotic resistance selection markers for: (1) maintaining the plasmid in *Escherichia coli* (*E. coli*) using ampicillin; and (2) confirming incorporation of the plasmid into the mammalian genome using Zeocin or neomycin, respectively.

Production of recombinant proteins from cell lines has become a routine procedure, and it can be achieved at either a large industrial or a smaller laboratory scale. Through this process, in-house production of proteins for a multitude of uses has become far more affordable and accessible using yeast, insect, plant and mammalian cells in suspension [11,12]. Additionally, protein-expression systems using baculoviruses and insect cells have been licensed in the USA since 2013 for the production of rHA-based vaccines such as Flublok, originally developed by Protein Sciences Corp. and now owned by Sanofi Pasteur (Paris, France) [6].

As described, mammalian cells have been used for expression, specifically, the human epithelial kidney (HEK) 293-derived cell line EXPI293F by Thermo Fisher Scientific (Waltham, MA, USA). This system has the benefit to obtain post-translation modifications (e.g., glycosylation) that are more similar to modifications on wild-type viral particles as compared to expression of the same proteins in yeast, insect and plant-based systems [13]. They also provide the benefit of being developed into stable cell lines that can be cryopreserved for long-term storage and subsequent expansion for future protein expression needs. Equivalent recombinant protein production systems based on other cell lines are also available, such as the Expi293F inducible cells [14]. However, this system would require the further addition of an inductor to the culture, e.g., tetracycline, for the initiation of the recombinant protein expression and thus affecting the overall production costs.

Once a protein is expressed, Immobilized Metal Affinity Chromatography (IMAC) can be used as a highly specific and efficient means of protein isolation from culture supernatants. Proteins purified through the IMAC method can yield very high purities, especially since few cellular proteins have a sufficiently strong affinity to remain associated with the immobilized metal matrix following a washing step with buffer containing a low concentration of imidazole. Due to the poly-histidine (His)-tag being of a relatively small size in comparison to the recombinant HA or NA proteins, and its location being at the C- or N-terminus of the protein, its incorporation into the protein sequence should not disrupt their native conformation or biologic activity of the protein. In this context, generation of physiologically relevant recombinant HA and NA proteins, such as in the case of other recombinant proteins, requires that the proteins remain soluble in the culture medium and that they maintain an equivalent conformation as the analogous proteins expressed by the corresponding wild-type influenza viruses. Likewise, recombinant HA and NA proteins will ideally maintain functional activity after purification, and even after prolonged storage. In the case of HA, which is responsible for viral attachment to cells of the upper respiratory tract [15], this entails binding of sialic acid and the ability to agglutinate erythrocytes. For NA, which is responsible for viral egress, this includes enzymatic activity and cleavage of terminal sialic acid moieties.

Herein, we detail an expression pipeline at key checkpoints that facilitated large-scale production of recombinant HA and NA proteins to high purity and with consistent yields across production runs. Moreover, we evaluated these proteins using a variety of laboratory techniques to assess their conformational structure, temperature stability and functional activity.

## 2. Materials and Methods

### 2.1. Plasmids

Plasmids for expressing recombinant wild-type HA influenza proteins were human codon optimized and ordered from Genewiz (South Plainfield, NJ, USA) or cloned from a full-length HA gene into a pcDNA3.1/Zeo (+) vector (Thermo Fisher Scientific) using the *BamH*I and *Hind*III restriction sites. Alternatively, HA possessing the Y98F mutation to attenuate sialic acid binding (HAΔSA) were also cloned into the pcDNA3.1/Zeo (+) vector [16]. The 3′ end of HA genes were truncated starting from amino acid 521 (H1N1 A/California/04/09 HA numbering) and modified to include a FoldOn of T4 fibritin trimerization domain to promote its expression as a soluble trimeric protein, as previously described [17]. An Avi-Tag sequence followed by a 6× His-Tag was also included downstream of the trimerization domain to enable site-specific biotinylation [18] and purification of the recombinant HA by IMAC, respectively (Figure 1A).

Plasmids encoding human codon optimized full-length influenza NA proteins were synthetically synthesized by Genewiz. The truncated NA genes were PCR amplified with *Kpn*I and *EcoR*I overhang primers into a pcDNA3.3-TOPO vector (Thermo Fisher Scientific). The resulting plasmids were sequence confirmed. Similar to previously described NA designed proteins [19], the gene for NA was truncated at the 5′ end starting from amino acid 75 (H3N2 A/Texas/50/12 NA numbering) removing the transmembrane domain and a portion of the stem, which was replaced with the tetrabrachion domain from *Staphylothermus marinus* and separated from the NA gene by three linker amino acids (GSG or GTG). A thrombin cleavage site was incorporated upstream of the tetrabrachion domain and preceded by a 6× His-Tag and lastly a CD5 signal sequence (Figure 1B).

For DNA plasmid amplification, chemically competent TOP10 or DH5α *E. coli* cells (Thermo Fisher Scientific) were used for bacterial transformation using 25–250 ng of the original DNA. *E. coli* chemically competent cells were transformed following the instructions provided by the manufacturer (Zymo Research, Irvine, CA, USA) and plasmids purified from a single transformed bacterial colony. In brief, *E. coli* transformed colonies were grown at 37 °C in Luria–Bertani (LB) broth with aeration by growing them in a shaker incubator set at 225 rpm. Ampicillin (100 µg/mL) and kanamycin (50 µg/mL) were used for antibiotic resistant selection depending on the plasmid. A nanodrop spectrophotometer (DeNovix, Wilmington, DE, USA) was used to quantify plasmid preparations and measure purity. A 260/280 absorbance ratio greater than 1.5 was considered acceptable for downstream steps.

DNA plasmids were digested using restriction enzymes (*BamH*I/*Hind*III for HA and *Kpn*I/*EcoR*I for NA) according to the manufacturer’s instructions (New England Biolabs, Ipswich, MA, USA) and then resolved by gel electrophoresis using O’GeneRuler 1 kb Plus DNA Ladder (Thermo Fisher Scientific) for reference to confirm their identity. SYBR safe DNA gel stain (Thermo Fisher Scientific) was incorporated into 1% *w/v* SeaKem LE agarose (Lonza, Basel, Switzerland) prior to casting and gels were run at 90–120 V until loading dye reached the end of the gel before imaging under UV light using the Chemi-Doc imaging system (Bio-Rad, Hercules, CA, USA).

### 2.2. Adherent Cell Culture

For adherent growth, EXPI293F cells (Thermo Fisher Scientific) were passaged at 70–90% confluency. Cells were dissociated using a trypsin-EDTA (0.05%) solution (Thermo Fisher Scientific), and then adjusted to 10^5^ cells/mL before seeding into Falcon 75 cm^2^ rectangular canted neck cell culture flasks (Corning, Corning, NY, USA) with vented cap with Dulbecco’s Modified Eagle Medium (DMEM, Thermo Fisher Scientific) supplemented with 10% fetal bovine serum (FBS, Atlanta Biologicals, Flowery Branch, GA, USA) and 1% penicillin–streptomycin (Thermo Fisher Scientific) in a 37 °C incubator with 5% CO_2_ and high humidity. For stable cell lines, medium containing Geneticin or Zeocin (at the appropriate concentration detailed in Section 2.6) was replenished every 3–5 days.

### 2.3. Suspension Cell Culture

For suspension cell culture, EXPI293F cells were maintained in suspension cultures in Expi Expression Medium (Thermo Fisher Scientific) at 37 °C, 8% CO_2_ and high humidity on a shake platform set to 125 rpm. As per manufacturer recommendation, cell density was maintained at 0.3–8 × 10^6^ cells/mL in polycarbonate vented Erlenmeyer flasks (Corning) containing a medium volume of 1/4–1/3 of the total volume of the flask. For stable cell lines, medium containing Geneticin or Zeocin (at the appropriate concentration detailed in Section 2.6) was replenished every 3–5 days.

### 2.4. Mouse B-Cell Hybridoma Cell Lines

SP2/0 mouse myeloma cell line (kindly provided by Dr. L. Wysocki, University of Colorado at Denver, Denver, CO, USA) and previously generated B-cell hybridomas (4H4, 4G10, 2A10 and 1F8) were cultured as already described [20]. In detail, mouse cell lines were maintained in B cell medium (BCM) consisting of RPMI 1640 medium (Sigma-Aldrich, Saint Louis, MO, USA) containing 10% FBS (Atlanta Biologicals), 23.8 mM sodium bicarbonate (Thermo Fisher Scientific, Waltham, MA, USA), 7.5 mM 4-(2-hydroxyethyl)-1-piperazineethanesulfonic acid (HEPES; Amresco, Solon, OH, USA), 170 mM penicillin G (Tokyo Chemical Industry, Tokyo, Japan), 137 mM streptomycin (Sigma-Aldrich), 50 mM 2-mercaptoethanol (Sigma-Aldrich), nonessential amino acids (Thermo Fisher Scientific) and 1 mM sodium pyruvate (Thermo Fisher Scientific). For monoclonal antibody (mAb) production, hybridoma cell lines were grown in BCM containing 2% Super Low IgG FBS (HyClone, Logan, UT, USA).

### 2.5. Transient Transfection of Suspension Cells

Transient transfection typically requires 10–15 days from culture expansion to purified protein (Figure 2). EXPI293F cultures with >95% viability were centrifuged and resuspended in fresh medium at 3 × 10^6^ cells/mL within 1 h prior to transfection. In this case, cells were transfected using the ExpiFectamine 293 Transfection Kit according to manufacturer’s instructions. DNA was diluted in 5% of final culture volume while in a separate conical tube, ExpiFectamine was diluted in 5% of the final culture volume to achieve a final culture concentration of 1 μg DNA/mL and 2.7 μL/mL, respectively. Diluted mixtures were incubated for 5 min at room temperature (RT). DNA mixture was added to ExpiFectamine mixture and incubated at RT for 20 min before addition to cells. Twenty hours post-transfection, 5% and 0.5% of final culture volumes were added of Enhancer 2 and Enhancer 1, respectively, as per manufacturer’s instructions. As an example, 500 mL of final culture volume requires 500 μg of DNA diluted in 25 mL of medium and 1350 μL of ExpiFectamine diluted in 25 mL of medium. After 20 h, 25 mL of Enhancer 2 and 2.5 mL of Enhancer 1 are added to transfected flasks. Protein expression in culture supernatant was confirmed 48–72 h post-transfection and harvested 7 days post-transfection or when viability reached <60%.

### 2.6. Transfection of Adherent Cells and Generation of Stable Cell Lines

HA and NA recombinant proteins were also expressed through the generation of stable transfected cells (Figure 2). EXPI293F cells were seeded in 6-well plates at a density of ~7 × 10^5^ cells/well. After 20–28 h, corresponding to a 60% cell confluency, they were transfected with the Lipofectamine 3000 (Thermo Fisher Scientific) transfection kit following the instructions provided by the manufacturer. Briefly, a mixture of 3 μg of DNA, 190 μL of Opti-Minimum Essential Medium (MEM) I (Thermo Fisher Scientific) and 6 μL of P3000 reagent was prepared. A second mixture of 6 μL Lipofectamine 3000 and 190 μL of Opti-MEM was prepared and both were incubated at RT for 5 min. The mixture containing DNA was added to the mixture of Lipofectamine, gently mixed and incubated an additional 5 min prior to addition to wells containing 1.5 mL of Opti-MEM I. Culture supernatants were collected 48–72 h post-transfection and assessed for protein expression. Following transfection, 6-well HA or NA transfected cells were supplemented with 100 μg/mL of Zeocin (InvivoGen, San Diego, CA, USA) or 250 μg/mL Geneticin (Thermo Fisher Scientific), respectively, based upon previous calculation of the minimal inhibitory concentration. Transfected cells were periodically expanded under drug selection and expression of recombinant HA or NA was periodically checked by Western blot as described below.

To select stable transfected cells expressing higher yields of rHA, a limiting dilution approach was performed in 96-well plates. In brief, 5 × 10^3^ cells were seeded in well A1 containing 150 μL selective growth medium and a 3-fold serial dilution of cells was performed vertically and a 2-fold dilution horizontally. Selective drug medium was replenished every 3–5 days until at least 5 wells having at a ~60% confluence and containing viable and a single subpopulation-derived cell colony were screened for expression through Western blot as described in Section 2.7. Wells showing the highest level of expression as per Western blot screening were expanded as adherent cultures (as described in Section 2.2) and were then re-adapted to grow in suspension by passaging and growing them in Expi Expression Medium (as detailed in Section 2.3) under drug selection.

### 2.7. SDS-PAGE and Western Blot

Culture supernatants containing secreted or purified proteins or influenza viruses, propagated in embryonated chicken eggs as previously described [21], were mixed with 4× Laemmli sample buffer (Bio-Rad) and subjected to sodium dodecyl sulfate-polyacrylamide gel electrophoresis (SDS-PAGE). In the same gel, a Spectra Multicolor Broad Range Protein Ladder (Thermo Fisher Scientific) was also included as a reference. Additionally, 250 ng of His-tagged rHA or rNA standards were also loaded in the same gel as positive controls for the Western blot and as references to evaluate the level of rHA and rNA expression, respectively. In brief, precast 10% SDS gels (Thermo Fisher Scientific) were electrophoresed at 200 V for 30 min. Proteins were transferred to polyvinylidene difluoride (PVDF) membranes using a Trans-Blot Turbo device (Bio-Rad) using the preset high molecular weight conditions and membranes were processed using the iBind device (Thermo Fisher Scientific) following the manufacturer’s instructions. Primary, mouse anti-His-Tag (clone J099B12, BioLegend, San Diego, CA, USA) at 1:1000, a group 1-specific mouse mAb (clone 15B7; Cat. No. IT-003-001M14, Immune Technology Corp., New York, NY, USA) or a group 2-specific mouse mAb (clone 34C9; Cat. No. IT-003-00423M13; Immune Technology Corp.) at 1:2000 and a secondary horseradish peroxidase (HRP)-conjugated goat-anti-mouse IgG (Southern Biotech, Birmingham, AL, USA) antibody at 1:4000 were used. Visualization was accomplished using Clarity ECL substrate (Bio-Rad) and membranes were imaged using the Chemi-Doc imaging system (Bio-Rad).

### 2.8. Protein Purification and Quantification

Cell debris and macrovesicles were removed from conditioned media by centrifugation at 10,000× *g* for 20 min at 4 °C in a fixed-angle rotor, followed by filtration through 0.22-μm PES membrane filter units (Corning). Recombinant His-tagged proteins were then purified by IMAC using HisTrap Excel columns (GE Healthcare, Chicago, IL, USA) and an automated purification ÄKTA Pure system (GE Healthcare). In brief, columns were first equilibrated with 5 column volumes of 20-mM NaPO_4_ and 0.5-M NaCl buffer using a flow rate of 1 mL/min/mL of bead resin. As per manufacturer’s recommendation, a flow rate of 1 mL/min/mL of bead resin was also used for sample application and all subsequent washing and elution steps. In greater detail, following completion of sample application, HisTrap Excel columns were washed with 20 column volumes of a 25 mM imidazole, 20 mM NaPO_4_ and 0.5 M NaCl buffer to remove non-specifically bound proteins. His-tagged recombinant HA or NA proteins were then eluted using 5 column volumes of a 250 mM imidazole, 20 mM NaPO_4_ and 0.5 M NaCl buffer and collected as 1 column volume (5 mL) fractions (Figure 3). Protein fractions were then concentrated and dialyzed against PBSA (phosphate buffered saline supplemented with 0.1% *w/v* NaN_3_) using Amicon 15 mL 30 K centrifugal concentrator (Corning) by spinning at 3000× *g* for 30 min. Protein samples were then quantified using a spectrophotometer by measuring absorbance at 260/280 nm and a bicinchoninic acid assay (Thermo Fisher Scientific). To assess purity, protein samples were run on SDS-PAGE for 30 min at 200 V and stained with Coomassie blue (Thermo Fisher Scientific). For experiments involving the use of previously described mAbs, their purification was performed as already described [20].

### 2.9. Thermal Treatment of Recombinant HA Protein

To assess the thermal stability of rHA protein, rHA from A/California/04/2009 (CA/09) was subjected to different temperature incubations (30–85 °C range) for 1 h using a MultiGene OptiMax Thermal Cycler (Labnet International, Edison, NJ, USA). The different treated rHA were then immediately used to coat ELISA plates as described below.

### 2.10. Enzyme-Linked Immunosorbent Assays (ELISA)

To assess the antigenicity of generated rHA, its trimerization and possible conformational changes due to different temperature incubations, enzyme-linked immunosorbent assays (ELISA) were performed as previously described [3,20,22]. In brief, Immulon 4HBX plates (Thermo Fisher Scientific) were coated overnight at 4 °C in a humidified chamber with 50 μL per well of a carbonate buffer solution (pH 9.4) containing 1 μg/mL of the native, the different temperature-treated in-house expressed and purified rHA from CA/09, as well as commercial trimeric (Protein Sciences, Meriden, CT, USA) and monomeric (Immune Technology) rHA from the same strain. For the thermal stability experiments, the murine mAb clones 15B7 (Immune Technology Corp.), 1F8 [20] and J099B12 (BioLegend) and the human mAb CR6261 (Creative Biolabs, Shirley, NY, USA) [23] were diluted in blocking buffer (2% bovine serum albumin [BSA] and 1% gelatin in PBS/0.05% Tween-20; all supplements from VWR International, Radnor, PA, USA) at a final concentration of 5 μg/mL. Alternatively, a CA/09-specific polyclonal mouse serum obtained from a previous study [3] was diluted 1:5000 in blocking buffer.

For the experiments assessing the trimeric conformation of the generated rHA, the murine mAb clones 2A10 [20] and AT171.718.57 (IRR) were 3-fold diluted starting from 100 μg/mL and used as primary antibodies. Binding of all primary antibodies was performed for 1 h at 37 °C. Plates were washed five times with PBS, 100 μL per well of HRP-conjugated goat anti-human or anti-mouse IgG (Southern Biotech) diluted 1:4000 in blocking buffer was added, and plates were incubated at 37 °C for 1 h. Finally, plates were washed five times with PBS and 2,2′-azino-bis(3-ethylbenzothiazoline-6-sulfonic acid) (ABTS) substrate (VWR International) was added. Plates were incubated at 37 °C for 15–20 min. Colorimetric conversion was terminated by addition of 1% SDS (50 μL per well), and OD was measured at 414 nm (OD414) using a spectrophotometer (PowerWave XS; BioTek, Winooski, VT, USA).

### 2.11. Hemagglutination Assay

The hemagglutination assay was used to assess functionality of rHA to agglutinate erythrocytes. Hemagglutination assays were performed similarly to previously described protocol [24]. In brief, 50 μL of duplicate two-fold serial dilutions of representative wild type rHA from A/Singapore/6/86 (Sing/86) and A/New Caledonia/20/99 (NC/99), their corresponding Y98F mutant versions (HAΔSA) and influenza viruses (positive controls), and BSA (negative control) were diluted in PBS, starting from 50 μg/mL for proteins or 5.25 × 10^8^ and 4.1 × 10^8^ plaque forming units per milliliter (PFU/mL) for Sing/86 and NC/99 influenza viruses, respectively, and incubated for 30 min with an equal volume of 0.8% turkey erythrocytes diluted in PBS. Erythrocytes were washed and used the day of the assay. The plates were mixed by agitation and covered, and the erythrocytes were settled for 30 min at RT. The hemagglutination titer was determined by the reciprocal dilution corresponding to the rHA concentration of the last well that contained agglutinated erythrocytes. Images of hemagglutination assays were acquired using the ImmunoSpot S6 ULTIMATE (Cellular Technology Limited, Shaker Heights, OH, USA).

### 2.12. Neuraminidase Activity Assay

Neuraminidase activity assays were performed to evaluate the sialidase activity of NA and similarly to a previous described protocol for the enzyme-linked lectin assay (ELLA) [25]. High affinity Immunoblot 4HBX 96-well flat-bottom plates (Thermo Fisher Scientific, Waltham, MA, USA) were coated overnight at 4 °C with 100 μL of 25 μg/mL fetuin (Sigma-Aldrich, St. Louis, MO, USA) in commercial KPL coating buffer (Seracare Life Sciences Inc, Milford, MA, USA). Purified recombinant NA proteins were diluted in sample diluent (PBS, 1% BSA, 0.5% Tween-20) to an initial concentration of 4 μg/mL. Before protein addition, fetuin plates were washed three times in PBS-T (PBS + 0.05% Tween-20). Then, 50 μL of two-fold serial dilutions of rNA protein were added to the fetuin-coated plate containing 50 μL of sample diluent in quadruplicate. A negative control column was included containing 100 μL of sample diluent only. Plates were sealed and incubated for 18 h at 37 °C and 5% CO_2_. After incubation, plates were washed six times in PBS-T, and 100 μL of HRP-conjugated peanut agglutinin (Sigma-Aldrich, St. Louis, MO, USA) diluted 500-fold in conjugate diluent (PBS, 1% BSA) was added. Plates were incubated at RT for 2 h. Plates were washed three times in PBS-T, and 100 μL (500 μg/mL) of o-phenylenediamine dihydrochloride (OPD; Sigma-Aldrich, St. Louis, MO, USA) in 0.05-M phosphate–citrate buffer with 0.03% sodium perborate pH 5.0 (Sigma-Aldrich, St. Louis, MO, USA) was added to the plates. Plates were immediately incubated in the dark for 10 min at RT. The reaction was stopped with 100 μL of 1 N sulfuric acid. The absorbance was read at 490 nm using a spectrophotometer (PowerWave XS; BioTek, Winooski, VT, USA).

### 2.13. rHA Fluorescent Probe Conjugation

Recombinant HA proteins possessing the Y98F mutation to attenuate sialic acid binding (HA∆SA) were dialyzed into 10 mM Tris-HCl (pH 8.0) and total protein concentration adjusted to ~2 mg/mL after BCA estimation. HA∆SA proteins were biotinylated with biotin protein ligase (BirA) (Avidity Biosciences, La Jolla, CA, USA) following manufacturer recommendations and confirmed by direct ELISA. High binding ELISA plates (CoStar, Washington, DC, USA) were coated overnight with 1 to 10 ng/well of BirA conjugated HA∆SA protein in carbonate buffer (pH 9.4) at 4 °C. ELISA plates were blocked and biotinylated HA∆SA proteins detected with HRP-conjugated streptavidin (Southern Biotech) followed by addition of the ABTS substrate. Biotinylated protein standard and a negative control protein were purchased from Avidity Biosciences (La Jolla, CA, USA; Cat. No. BIS-300). Fluorescently-conjugated HA∆SA probes were generated by stepwise addition of high concentration (1 mg/mL) PE- or APC-conjugated streptavidin (SA-PE or SA-APC) (BioLegend) to a final 8-fold molar excess, in ice-cold PBS with Protease Inhibitor Cocktail Set I (Calbiochem, San Diego, CA, USA; Cat. No. 539131). HA∆SA probes were stored at 4 °C in the dark up to two days.

### 2.14. Flow Cytometry Probe Staining

The previously characterized HA-specific 4G10 hybridoma cell line [20] (2 × 10^6^) was stained with 250 ng of APC conjugated rHAΔSA probes in 200 μL of staining buffer (PBS/2% FBS) for 30 min on ice and protected from light. Cells were then washed by centrifugation, resuspended in 100 μL of staining buffer, and 5 μL of 7-aminoactinomycin (7-AAD) live/dead staining was added 5 min prior to acquisition in a LSRII flow cytometry analyzer (BD Biosciences, San Jose, CA, USA). Cell surface BCR expression was evaluated by staining with APC-conjugated goat anti-mouse IgG at 1:100 (BioLegend, Cat. No. 405308). The SP2/0 myeloma and the HA-negative 4H4 hybridoma cell lines were similarly stained and used as negative controls.

Alternatively, 5 × 10^6^ of banked cryopreserved peripheral blood mononuclear cells (PBMCs) from a previously identified highly reactive influenza vaccinated individual [26], collected 7 days post vaccination, were first treated with Fc receptor blocking solution (BioLegend; Cat. No. 422301) and then simultaneously stained with 250 ng of phycoerythrin (PE)-conjugated CA/09 HAΔSA and 250 ng allophycocyanin (APC)-conjugated A/Hong Kong/4108/2014 (HK/14) HAΔSA. Excess unbound HA∆SA probes were washed by centrifugation and human PBMC were then stained with an antibody cocktail for B cell immunoprofiling as previously described [26]. Data were acquired on a FACSAria Fusion flow cytometer (BD Biosciences) and analyzed using the FlowJo10 software (BD, Franklin Lakes, NJ, USA).

### 2.15. Statistical Analysis

All graphically represented results are reported as absolute mean values plus standard deviations. Differences in binding of antibodies to rHA treated at different temperatures as compared to the reference rHA were analyzed using the ordinary one-way ANOVA test. All statistical analysis was performed using Prism V.8.4.2 software (GraphPad, San Diego, CA, USA), and a *p* value lower than 0.05 was considered statistically significant.

## 3. Results

### 3.1. Expression and Purification of Recombinant HA and NA

As wild-type HA and NA are trimeric and tetrameric protein complexes, we incorporated a trimerization or a tetramerization domain, respectively, to promote and stabilize their oligomerization [17] (Figure 1). As shown in Figure 4A, following purification and Coomassie staining of SDS-PAGE loaded proteins, the monomer fraction of rHA constitutes the major species, with a size of ~90 kDa (red arrow). Lower intensity bands are also visible at ~180 and ~270 kDa, and represent rHA oligomers, dimers (yellow arrow) and trimers (white arrow), respectively. Another lower intensity band is visible at ~65 kDa, possibly representing a less glycosylated rHA monomer isoform (blue arrow). Additionally, the migration pattern of rHA is similar to HA derived from influenza viral particles, as shown in the Western blot depicted in Figure 5.

The major bands of rNA are ~150 (green arrow) and ~300 kDa (orange arrow), corresponding to rNA dimers and tetramers, respectively. A low intensity band is also visible at a 75 kDa approximate size (pink arrow), corresponding to the rNA monomer. On average, the yields from rHA or rNA stable transfected cell line ranged nearly 20-fold (1.5–28 mg rHA/L). As shown in Figure 4B, the maximal rHA yields were achieved by the previously described Computationally-Optimized Broadly Reactive Antigen (COBRA) P1 HA protein [3,24] stable transfected cells. Variable, but consistent, rHA yields were achieved with CA/09, A/Brisbane/02/2018 (Brisb/18), A/Switzerland/9715293/2013 (Switz/13) and HK/14 rHA stable transfected cell lines. Unfortunately, despite undergoing Zeocin selection and limiting dilution, the Sing/16 rHA stably transfected line remained a poor producer. Importantly, consistent recombinant protein expression levels were maintained with all the stable cell lines, even after approximately five months of passaging (the longest time period we were able to test) following their generation (data not shown).

As an additional means of characterizing the purified rHA, their size and migration patterns following SDS-PAGE were further characterized through Western blot analysis. In comparison to native HA present on the surface of influenza virions, the corresponding rHA demonstrated a comparable composition (Figure 5A). Furthermore, as depicted in Figure 5B–D, purified rHA were specifically recognized by group 1- and group 2-specific mAbs for H1N1 and H3N2 rHA, respectively.

### 3.2. Recombinant HA Are Endowed of Functional Activity In Vitro

To assess their functional activity and capacity to agglutinate red blood cells (RBCs), rHA proteins and the corresponding H1N1 or H3N2 influenza virus were evaluated in parallel. As shown in Figure 6, rHA from Sing/86 and NC/99 exhibited hemagglutination activity, with minimal effective concentrations of 0.1 and 0.8 μg/mL, respectively. In agreement with previous reports [17,28], Sing/86 and NC/99 rHA proteins possessing a Y98F mutation (HAΔSA), along with the BSA negative control, lacked detectable hemagglutination activity. Accordingly, the corresponding Sing/86 and NC/99 influenza viruses possessed hemagglutination activity at a minimal viral titer of 10^6^ and 5 × 10^6^ PFU/mL, respectively.

### 3.3. Thermal Stability of Recombinant HA

An important consideration and logistical barrier to mass vaccination efforts against seasonal influenza and other pathogens, especially in under-developed countries in which maintenance of cold-chain may not be feasible, is vaccine stability and immunogenicity. To this end, we sought to evaluate the thermal stability, and specifically preservation of conformational epitopes, of representative recombinant HA (CA/09) proteins following incubation at increasing temperatures. Specifically, following incubation at increasing temperatures (30–85 °C), the CA/09 rHA protein was used to coat wells of an ELISA plate and probed with either polyclonal or mAb preparations to monitor the structural integrity of conformational epitopes. As shown in Figure 7, binding of conformation-sensitive mAbs (CR6261 and 1F8) and polyclonal serum is higher or similar to that achieved to the reference rHA, at temperatures ranging from 30 to 54 °C. Higher temperatures (60–85 °C) caused a linear decrease of the binding activity of polyclonal serum and of HA stem- and head-directed conformation-sensitive mAbs (Figure 7C,D). Interestingly, the binding of mAbs directed to an HA stem linear epitope (15B7) or to the His-Tag (J099B12) was also affected by the different temperature incubations (Figure 7A,B). However, differently from conformation-sensitive mAbs and polyclonal serum, their magnitude of binding was not linearly affected by the different temperature treatments.

### 3.4. Recombinant NA Proteins Are Endowed with Enzymatic Activity In Vitro

To assess the functional activity of purified NA recombinant proteins, a neuraminidase activity assay was performed to evaluate their ability to cleave fetuin-borne sialic acid.

As reported in Figure 8, representative proteins from H1N1 and H3N2 strains are endowed with neuraminidase activity, as demonstrated by the NA dose-dependent increased specific binding of peanut agglutinin to the sialic acid-cleaved fetuin protein. Interestingly, the order of NA activity featured by NA recombinant proteins is similar to those exerted by the corresponding NA on the influenza viral particles.

In particular, H1N1 recombinant NA from A/Brisbane/59/2007 (Brisb/07) and A/Texas/36/1991 (TX/91) had an EC_50_ of 50 ng/mL. Alternatively, H1N1 A/Chile/1/83 (Chile/83) and Brisb/18 showed an EC_50_ of 20 ng/mL. The H1N1 CA/09 NA had the lowest EC_50_ corresponding to 10 ng/mL. Differently, H3N2 recombinant NA from TX/12 and Switz/13 showed an EC_50_ of 40 ng/mL. On the other hand, the one from HK/14 showed an EC_50_ of 65 ng/mL.

### 3.5. In-House Produced Recombinant HA Are Trimerized

To confirm the FoldOn-driven trimerization of rHA, an ELISA was performed. In detail, a representative in-house expressed and purified rHA, a trimeric recombinant H1N1 protein component of the commercial influenza Flublok vaccine formerly by Protein Sciences (now Sanofi-Pasteur) and a commercial recombinant HA monomer were probed with rHA-specific mAbs. As shown in Figure 9, the 2A10 mAb, whose binding activity is specific for the trimeric form of the HA [20], binds in a dose-dependent manner to either the in-house rHA or the commercial vaccine-derived trimeric rHA. No binding activity was confirmed against the HA monomeric rHA. Differently, the AT171.718.57 mAb clone, whose binding activity is specific for the monomeric form of the HA, shows a dose-dependent binding activity against the rHA monomer and is not able to bind either the in-house rHA or the commercial vaccine-derived trimeric rHA.

### 3.6. In-House Produced rHAΔSA Protein Enable Identification of HA-Specific B Cells

Identification of HA-specific B cells using rHA proteins possessing the Y98F mutation has been reported previously [17]. To demonstrate that our in-house expressed rHAΔSA proteins could also be used for similar purposes, we utilized an HA-reactive murine B cell hybridoma that have been reported previously [20]. The 4G10 hybridoma line expressed a comparable level of surface IgG to another hybridoma line (4H4) known to be not specific for HA, and thus were utilized as a simplified model for assessing the specificity of rHAΔSA staining. In accordance with previously published data for the corresponding 4G10 mAb [20], the 4G10 hybridoma cells stained positive with both the fluorescently-labeled COBRA P1 and CA/09 rHAΔSA reagent (Figure 10A). In contrast, the 4H4 hybridoma cell line failed to yield definitive staining with the same probes (Figure 10A). Collectively, the observed rHAΔSA reagent staining patterns support their usage for identification of HA-specific B cells in more complex samples, such as human PBMCs.

To further validate the use of rHAΔSA probes for the identification of HA-specific B cells, PBMCs from a donor that mounted a broadly-reactive response following seasonal influenza vaccination were stained with rHAΔSA probes representing the respective H1N1 and H3N2 vaccine strains. As expected, a well-defined plasmablast population (CD27^+^ CD38^++^) was present seven days after vaccination, and a subset of these cells were labeled with one or more rHAΔSA probes. Similarly, 6% of the CD27^+^ IgD^−^ B cell compartment (class-switched memory B-cells) were also labeled by the fluorescently-labeled rHAΔSA probes (Figure 10B). Importantly, the overall frequency of rHAΔSA-labeled cells was higher for both B-cells compartments in high vaccine-reactive subjects as compared to low vaccine reactive subjects (Figure 10C).

## 4. Discussion

Recombinant protein production, through the processes of cell culture transfection and stable cell lines, remains a viable technique for increasing yields of distinct proteins that are used as tools in a wide variety of applications, but mainly as antigens for diagnostic and prophylactic purposes [29]. Production of large protein quantities can also be achieved, while also moderating the overall cost, in comparison to purchasing similar commercially available proteins on the open market. Furthermore, the creation of stable cell lines reduces the budget and time required to perform multiple large-scale transfections and allows for a more consistent product over multiple runs.

This protocol is a set of guidelines for production and purification of soluble recombinant proteins through transient transfection and stable cell line production in mammalian cells from the initial plasmid creation to the final isolation step via IMAC. Through these means, we are able to create a product of high purity, as well as, high levels of consistency between runs. In addition, due to being synthesized in mammalian human-derived cells, the proteins have more similar glycosylation and functional activity to the native proteins expressed on the surface of the influenza virions as compared to non-mammalian cells [30]. As demonstrated by hemagglutination and neuraminidase assays, the produced rHA and rNA maintain similar functional activity as their native counterparts displayed on the surface of influenza viral particles. Additionally, crystallography study of COBRA-2 rHA protein generated through this expression pipeline was structurally preserved at the atomic level [31]. Likewise, both human or mouse mAbs recognizing conformational epitopes exhibit strong binding of rHA proteins isolated using similar methods [3,20,32].

Owing to their stability and structural integrity, the HAΔSA proteins serve as an excellent tool for specific labeling of HA-reactive B cells. This was evident through labeling an HA-specific B cell hybridoma (4G10) [20] as well as plasmablasts and memory B cells present within a responder following influenza vaccination. In fact, the use of recombinant antigens as “probes” for capturing antigen-specific B cells has important implications not only in an antibody discovery setting [33], but also in the context of investigating defined antigen-specific B cell populations (e.g., plasmablasts and memory B cells) following infection or vaccination [26,34].

The native conformation of our in-house produced rHA was largely maintained despite exposure to elevated temperatures. In fact, binding of mAbs recognizing conformational epitopes was generally preserved across a temperature range of 30–54 °C relative to rHA that was unmanipulated. However, a linear decrease in binding for these same mAbs was observed following treatment of the rHA at higher temperatures (60–85 °C), suggesting a disruption of the corresponding conformational epitopes. Interestingly, the binding of mAbs directed against a linear epitope or the His affinity tag were also affected by the temperature changes. However, alteration in binding signal did not follow a pattern, rather suggesting that increasing the temperature of rHA may have induced molecular dynamics and consequent modifications in the display of the relevant epitopes [35]. Additionally, as previously described, rHA conformational changes do occur following a low pH buffer treatment, as demonstrated by the generally decreased affinity of a panel of mAbs against the treated protein as compared to the native rHA [20].

Introducing a series of quality control steps into the expression pipeline, such a Western blot of culture supernatants, along with sequence analysis of plasmids and stable cell lines, serve as important checkpoints to ensuring the identity of the expressed and purified product. In this regard, we utilized commercially available mAb reagents to confirm the proteins expected size and reactivity with subtype-specific mAb probes (Figure 5B–D). Of additional importance, when multiple proteins are expressed and purified at the same time, during protein isolation via IMAC, it is also important to keep all protein supernatants free of cross contamination due to the common His-Tag found on all of our cell-line produced proteins. If cross contamination were to occur anywhere during our protocol, it would become nearly impossible to discern between proteins of the same subtype via these Western blot analyses. Instead, proper identification of any contaminating protein would require sequencing using mass spectrometry. For this reason, we encourage the use of protein-specific columns to avoid potential carry-over between successive purification.

While at this time our protocol is solely being used for recombinant protein production, transfection of suspension EXPI293F cells can be used for additional applications. As an example, we have confirmed that it may also be used for more affordable and larger scale virus-like particles (VLPs) and mAb production. This can be achieved through the transfection of the mammalian cells with multiple plasmids simultaneously, such as those encoding for the scaffold and envelope proteins or antibody heavy and light chains [36], respectively, yielding cells expressing multiple viral proteins and a higher yield as compared to adherent cell lines [37].

Additionally, our protocol is well-suited for scalability. In fact, through the use of multiple chromatography columns plugged to different or multi-channel peristaltic pumps, it is possible to run different conditioned media for the simultaneous purification of distinct recombinant proteins.

The availability of an expression platform endowed of a high yield capacity accompanied by an accurate recombinant protein expression as compared to the native counterpart is fundamental not only for the common practice in research laboratories, but also in a company setting aimed at producing large amounts of good manufacturing practices (GMP) quality-grade recombinant proteins as vaccine formulations or as high-quality antigens to be used for diagnostic purposes [38].

As a further example, the recombinant proteins generated through this described protocol have been extensively utilized in a pre-clinical setting to evaluate the breadth of the antibody response following immunization with universal influenza vaccine candidates [3,20]. In particular, the breadth of recognition of HA-specific B cells as well as polyclonal antibodies and mAbs, has been assessed by using rHA as antigens in immunological assays, such as ELISA and ELISpot.

Likewise, HAΔSA probes can be used to for the isolation of HA-reactive plasmablasts and memory B cells using Fluorescence-Activated Cell Sorting (FACS) [39]. Current studies are aimed at evaluating their antigenicity and immunogenicity, as sole recombinant proteins or conjugated to nanoparticles, for the pre-clinical evaluation of universal influenza vaccine candidates based on the main influenza virus surface glycoproteins [40]. However, the here described pipeline can be easily adopted not only to produce a wide variety of other influenza virus proteins (such as M2 and NP) [41], but also as an affordable, rapid and scalable methodology for recombinant protein-based vaccine or diagnostic platforms [42]. In this regard, this protein production pipeline has been easily converted by our group for the rapid large-scale production of SARS-CoV-2 recombinant proteins [43,44].

## 5. Conclusions

Collectively, the proteins generated through the expression pipeline detailed here are intended not only to serve as core reagents for the assessment of antibody breadth and durability elicited by universal influenza vaccine candidates, but also as prototype immunogens for next generation vaccines based on recombinant proteins.

## Figures and Tables

**Figure 1 vaccines-08-00462-f001:**
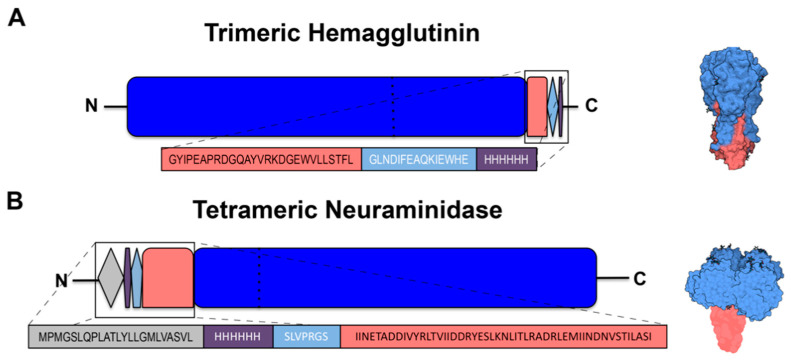
(**A**) Schematic of the design of the recombinant trimeric hemagglutinin. The N-terminus of the truncated HA protein (blue) is modified by the FoldOn trimerization domain (salmon) followed by the Avi-Tag (light blue) and 6× His-Tag (violet) regions. Separating dotted line inside HA indicates the HA1 head region (N-terminus) versus the HA2 stem region (C-terminus) also depicted in light blue and salmon, respectively, on the 3D drawing on the right. (**B**) Schematic of the design of the recombinant tetrameric neuraminidase. The CD5 signal sequence (grey) is at the N-terminus of the protein, followed by the 6× His-Tag (violet) and the thrombin cleavage site (light blue) sequences, which are followed by the tetrabrachion domain (salmon) linked to the stem region of the truncated neuraminidase protein (blue). Dotted line indicates the head region (C-terminus) and stem region (N-terminus) also depicted in light blue and salmon, respectively, on the 3D drawing on the right. Figure generated with Biorender.

**Figure 2 vaccines-08-00462-f002:**
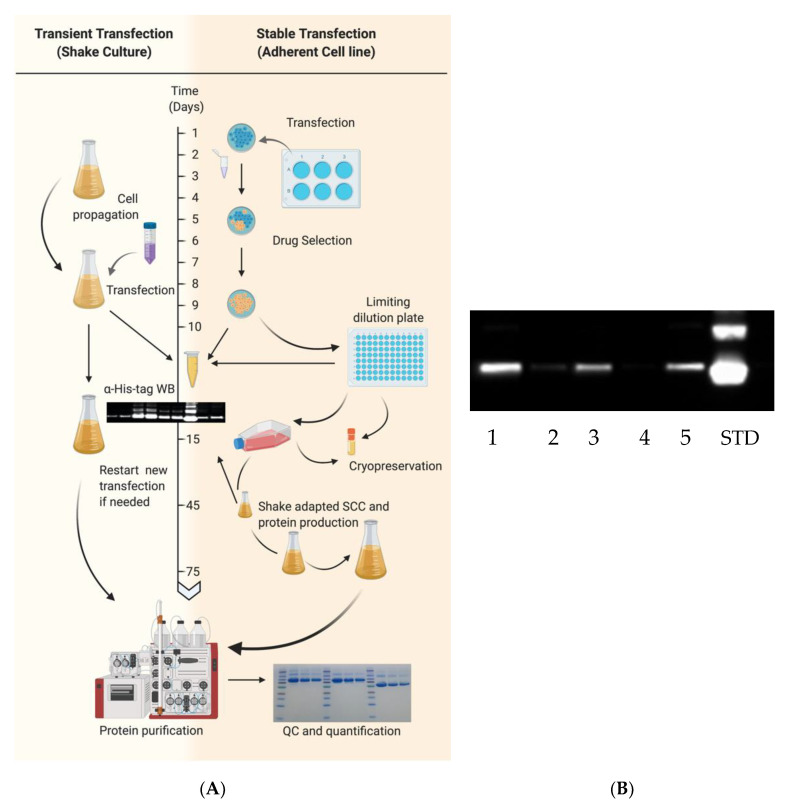
(**A**) Comparative timeline for the generation of transient and stable protein production cell lines. Transient transfection typically requires expansion of EXPI293F cells in suspension culture prior to large scale transfection. High density cultures of EXPI293F cells with viabilities exceeding 95% are pelleted by centrifugation, resuspended in serum-free media and transfected with ExpiFectamine 293 transfection kit, as described in the Materials and Methods Section. Efficient transfection and protein production are confirmed 48–72 h post-transfection by Western blot, and cultures are harvested 5–7 days post-transfection or when cell viability drops below 60%. Cultures are centrifuged and supernatant filtered to remove cellular debris before IMAC protein purification. Alternatively, generation of stable protein expressing cell lines began following a small-scale transfection in 6-well plates using adherent EXPI293F cells at ~60% confluency. In this case, cells are transfected with plasmid DNA using the Lipofectamine 3000 transfection kit as described in the Materials and Methods Section. Efficient transfection and protein production are confirmed 48–72 h by Western blot and cultures are placed under antibiotic selection. Subclones exhibiting high protein expression levels are then identified following limiting dilution and then expanded for cryopreservation. Subsequently, protein producing adherent cell lines are adapted to suspension growth in serum free Expi293F expression medium. Culture supernatants are then centrifuged and filtered prior to purification using the GE ÄKTA pure system. Following purification, proteins are quantified and Coomassie stained to assess purity. (**B**) Representative Western blot for assessing rHA expression levels from supernatants of limiting diluted stably transfected cell line subclones (numbered 1–5) from H3N2 A/Texas/50/2012 (TX/12). The last lane was loaded with 250 ng of His-tagged rHA from H1N1 A/California/04/2009 (CA/09) used as a standard. rHA were detected using an anti-His mAb as described in the Materials and Methods Section. Part of the figure was generated with Biorender.

**Figure 3 vaccines-08-00462-f003:**
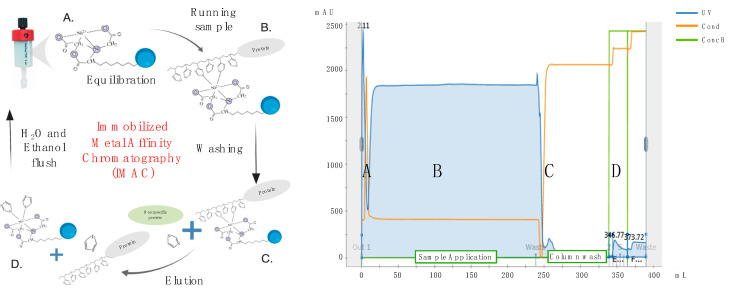
The left panel depicts the free and bound components at each stage of the Immobilized Metal Affinity Chromatography (IMAC) process. The right panel shows the UV readings yielded by the GE ÄKTA pure system with each letter representing the same stage of the purification as in the left panel. Part of the figure was generated with Biorender.

**Figure 4 vaccines-08-00462-f004:**
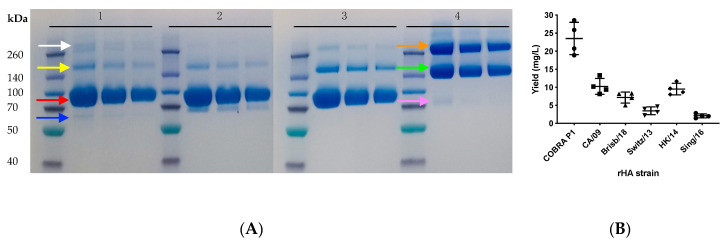
Representative Coomassie stained SDS-PAGE of purified recombinant proteins expressed by stably-transfected cell lines. From left to right in 1, 2, 3 and 4 Spectra Multicolor Broad Range Protein Ladder (Thermo Fisher Scientific), 8, 5 and 3 μg of purified rHA from H3N2 A/Singapore/INFIMH-16-0019/2016 (Sing/16), H1N1 A/New Caledonia/20/99, H1N1 COBRA P1 and rNA from H3N2 TX/12, respectively. For rHA, arrows indicate monomers (red), dimers (yellow) and trimers (white) and a less glycosylated rHA monomer isoform (blue). For rNA, arrows indicate dimers (green), tetramers (orange) and monomers (pink) (**A**). Representative yields from culture media of transfected EXPI293F cell lines stably expressing rHA proteins from H1N1 COBRA P1, CA/09, Brisb/18 and H3N2 Switz/13, HK/14 and Sing/16 (**B**).

**Figure 5 vaccines-08-00462-f005:**
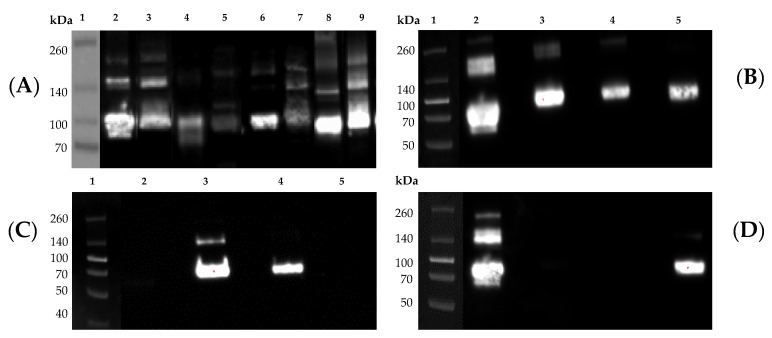
Representative western blot images of purified recombinant proteins expressed by stably-transfected cell lines and corresponding influenza virus HA and NA. (**A**) In lane 1: Spectra Multicolor Broad Range Protein Ladder (Thermo Fisher Scientific); Lane 2: COBRA P1 rHA (3 μg); Lane 3: COBRA P1 virus (6.7 × 10^6^ PFU); Lane 4: CA/09 rHA (0.5 μg); Lane 5: CA/09 virus (3.3 × 10^3^ PFU); Lane 6: COBRA X3 rHA (0.5 μg); Lane 7: COBRA X3 virus (6.7 × 10^5^ PFU); Lane 8: COBRA X6 rHA (0.5 μg); Lane 9: COBRA X6 virus (6.7 × 10^5^ PFU). The group 1-specific mAb (clone 15B7) was used as primary antibody. (**B**–**D**) Lane 1: Spectra Multicolor Broad Range Protein Ladder (Thermo Fisher Scientific); Lane 2: CA/09 rHA (1 μg); Lane 3: Sing/16 rHA (1 μg) Lane 4: H3N2 COBRA T10 rHA [27] (1 μg); and Lane 5: NC/99 rHA (1 μg) detected with: (**B**) anti-His-Tag (J099B12); (**C**) anti-group 2 (34C9); and (**D**) anti-group 1 (15B7) primary mAbs.

**Figure 6 vaccines-08-00462-f006:**
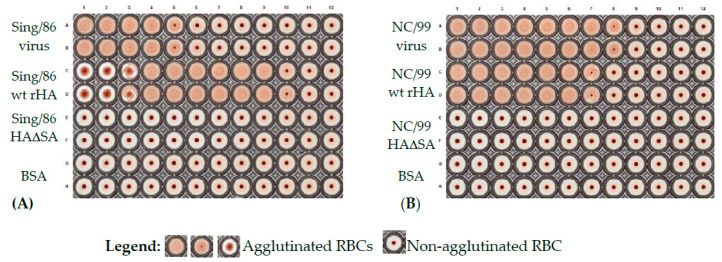
Hemagglutination activity of rHA expressed by stably-transfected cell lines. Representative wild type rHA and corresponding HAΔSA from Sing/86 (**A**) and NC/99 (**B**) were evaluated for their ability to agglutinate turkey RBCs. All proteins were tested in duplicate starting from 50 μg/mL and two-fold serially diluted (from left to right). BSA was used as negative control and the corresponding influenza viruses from Sing/86 and NC/99 were used as positive controls.

**Figure 7 vaccines-08-00462-f007:**
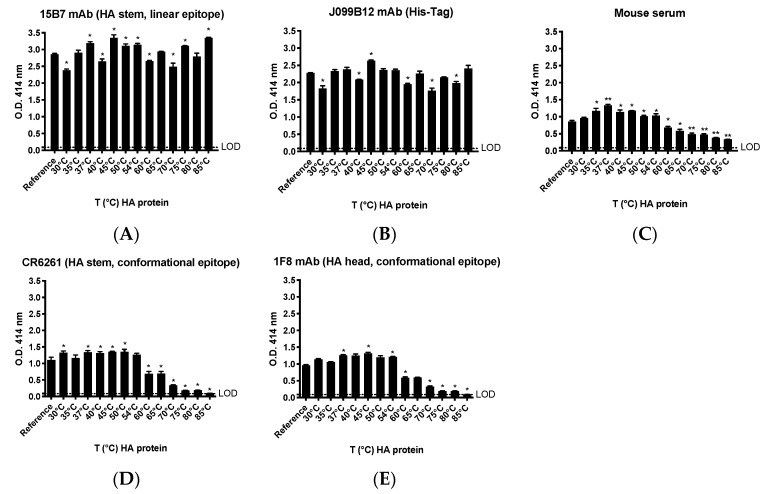
Thermal stability of stably-transfected cell line-derived purified rHA (from CA/09) following treatment at different incubation temperatures. After temperature treatments, the following was assessed: (**A**,**B**) binding of mAbs 15B7 (HA linear epitope) and J099B12 (His-Tag); (**C**) mouse CA/09-specific polyclonal serum; and (**D**,**E**) conformation-sensitive directed mAbs (CR6261 and 1F8). LOD represents the limit of detection. Asterisk on the top of bars indicates a statistically significant OD414 difference as compared to the reference protein (* *p* < 0.05; ** *p* < 0.01).

**Figure 8 vaccines-08-00462-f008:**
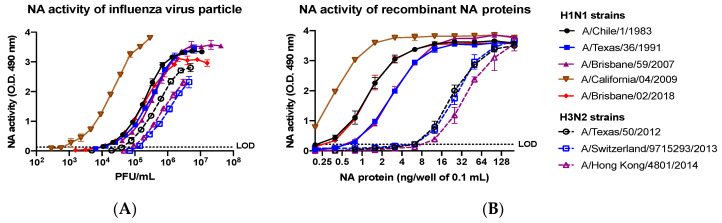
Neuraminidase activity of: wild-type influenza A virus (**A**); and the corresponding recombinantly expressed and purified NA proteins from stably-transfected cell lines (**B**). Recombinant NA proteins from H1N1 strains and H3N2 strains were evaluated for neuraminidase activity measured as the ability to cleave sialic acid displayed on the fetuin glycoprotein. LOD represents the limit of detection.

**Figure 9 vaccines-08-00462-f009:**
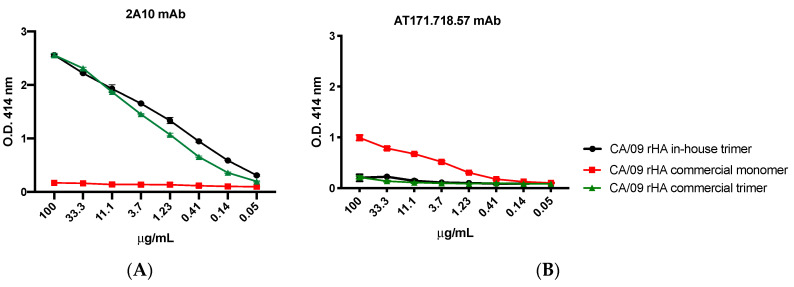
Binding activity of: murine mAbs specific for the HA trimer (clone 2A10) (**A**); and the HA monomer (clone AT171.718.57) (**B**). Commercial (rHA trimer from Protein Sciences and rHA monomer from Immune Technology Corp.) and in-house produced recombinant HA proteins from H1N1 CA/09 were used to evaluate the binding activity of the murine mAb clones 2A10 and AT171.718.57 at three-fold serial dilutions.

**Figure 10 vaccines-08-00462-f010:**
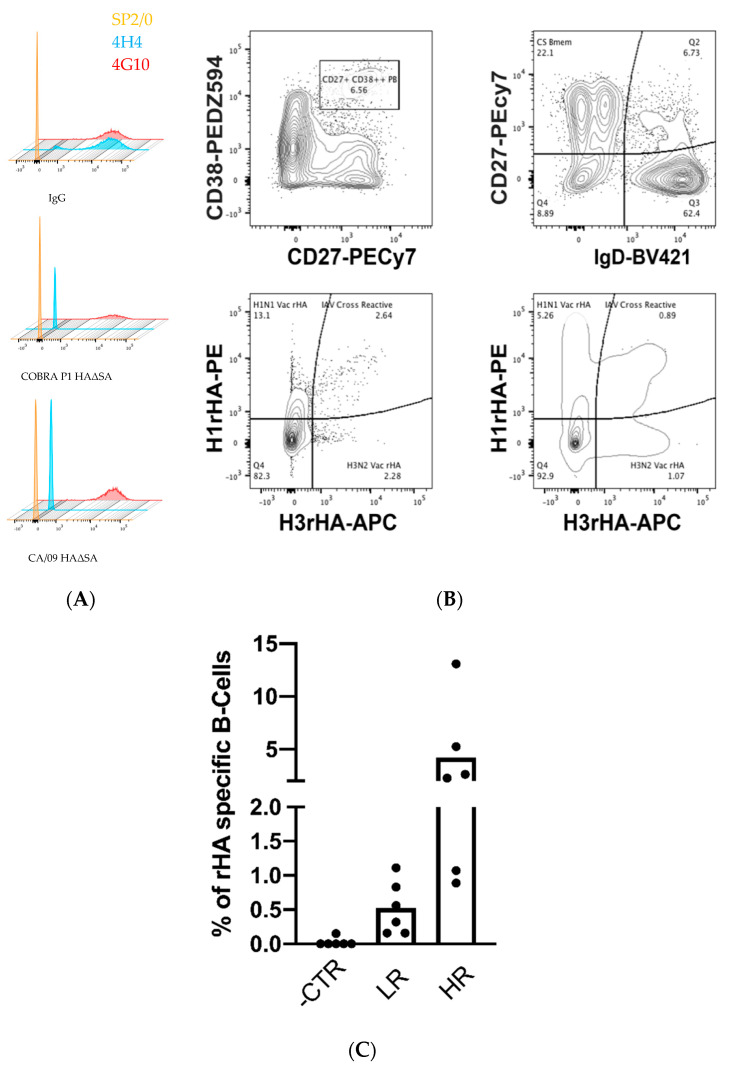
Fluorescent-conjugated rHAΔSA expressed by stably-transfected cell lines efficiently identify HA-specific B-cells. (**A**) Representative flow panels of HA-directed (4G10) H1N1 specific hybridoma stained with CA/09 and COBRA P1 rHAΔSA APC-conjugated probe. The SP2/0 cell line was used as negative control for IgG and rHAΔSA staining while the IgG^+^ 4H4 hybridoma cell line was used as negative control for rHAΔSA staining. (**B**) Representative flow panel of rHA-specific plasmablasts (top left) and class-switched memory B-cells (top right) in a highly reactive influenza vaccinated subjects following rHAΔSA H1 (*y* axis) and H3 (*x* axis) probe staining (bottom left and right) as described in the Materials and Methods Section. (**C**) Frequency of rHAΔSA-specific cells in different B-cell compartments, measured as in B, from low- (LR) and high- (HR) reactive subjects. Fluorescent minus one (FMO) controls were used to set positive staining gates.

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
