# Peer review of "High-Yield Expression and Purification of Recombinant Influenza Virus Proteins from Stably-Transfected Mammalian Cell Lines"

_vaccines, 2020, doi:10.3390/vaccines8030462_

Round 1

Reviewer 1 Report

Large scale protein production is needed for both in vitro and in vivo vaccine development. Several methods and stable cell lines approaches have been widely used in the field for protein production such as lentivirus system, inducible 293 cells, Flp-In system for expression of HA. In this work by Ecker et al, describe an approach to generate stable cell lines using EXPI293F cells and protein production pipeline for expression of recombinant proteins, influenza virus hemagglutinin (HA) and recombinant influenza virus neuraminidase (NA) proteins. Overall, authors present nice characterization of proteins produced through this method but is missing critical details of their strategy and would benefit from more explanatory details as listed in the following comments.

  1. As this work focusses on describing the method on stable expression of protein from cell lines, some of the critical details are missing. For example, in section 2.6 on generation of stable cell lines, how often the transfected cells are replenished with antibiotics? How are the colonies screened for protein expression, please include the critical steps to consider for western blotting? How many colonies are screened? Authors mention ‘”the wells with highest expression are expanded as adherent culture and were then re-adapted to grow”- describe the details. It would be also be preferred to include schematic with all the steps.
  2. Once the stable expressing cells are identified and cryopreserved, how long the cell lines last in culture. How many cells are needed for x amount of protein production? Can these stable cell lines be continuously passaged? Has authors confirm the expression of proteins from passage 1 to passage 10. This is critical information to help readers determining the benefits of generating these stable cell lines versus transient transection or other available tools. Authors should also include western blot confirming the expression of their clonal cell line for their selection step.
  3. Authors should clarify how is the protein yield from their method compared to other methods used or commercially available ones.
  4. Can authors briefly comment about using their strategy for protein production versus some of the other already in the available, such as lentivirus system, inducible 293 cells.
  5. This work reads more of method paper than a research article, so authors should include a detailed schematic of expression pipeline, highlighting the critical check points that will help users to adapt this method.

Minor comments:

  1. In figure 1, the color scheme described in figure legend is mixed up. In Figure 1A, its not clear what is referred to Foldon trimerization, what does “grey region” represents. In Figure 1B, Histidine is purple but marked yellow in figure legend.
  2. Some of the figure labels are not legible, for example Figure 10.
  3. Lines 534-535, include the Figure reference.

Author Response

Dear Reviewer,

First of all, we would like to thank you for your valuable comments that contributed to improve the overall quality of our manuscript.

Please below you can find a point by point response (highlighted in bold) to your comments. We hope you will find them satisfying.

“As this work focusses on describing the method on stable expression of protein from cell lines, some of the critical details are missing.

For example, in section 2.6 on generation of stable cell lines, how often the transfected cells are replenished with antibiotics? “

The technical details on this point have been added (lines 170-171 and 177-178).

“How are the colonies screened for protein expression, please include the critical steps to consider for western blotting? How many colonies are screened? Authors mention ‘”the wells with highest expression are expanded as adherent culture and were then re-adapted to grow”- describe the details.”

The technical details on these points have been added (lines 241-246).

“It would be also be preferred to include schematic with all the steps”.

All the steps for the generation of stable cell lines are illustrated in figure 2A which has been modified to include further details.

“Once the stable expressing cells are identified and cryopreserved, how long the cell lines last in culture. How many cells are needed for x amount of protein production? Can these stable cell lines be continuously passaged? Has authors confirm the expression of proteins from passage 1 to passage 10”.

The technical details on these points have been added (lines 463-466).

“Authors should also include western blot confirming the expression of their clonal cell line for their selection step”.

A representative western blot for confirming the expression of different subclones has been included in figure 2B.

“Authors should clarify how is the protein yield from their method compared to other methods used or commercially available ones.

Can authors briefly comment about using their strategy for protein production versus some of the other already in the available, such as lentivirus system, inducible 293 cells”.

Expression of rHA and rNA through the system we described is very variable not only at the stably-transfected but also at the transiently-transfected cell line level as shown in figure 4B. This variability strictly depends on the rHA/rNA strain under consideration. As suggested by the reviewer, comments and comparisons with other expression systems have been included in the introduction (lines 54-59; 72-75; 82-85).

“This work reads more of method paper than a research article, so authors should include a detailed schematic of expression pipeline, highlighting the critical check points that will help users to adapt this method”.

The schematic of the expression pipeline, including the critical check points are listed in figure 2A.

Minor comments:

“In figure 1, the color scheme described in figure legend is mixed up. In Figure 1A, its not clear what is referred to Foldon trimerization, what does “grey region” represents. In Figure 1B, Histidine is purple but marked yellow in figure legend”.

We thank the reviewer for having noticed that. Figure 1 has been modified accordingly.

“Some of the figure labels are not legible, for example Figure 10”.

We thank the reviewer for having noticed that. Figure 10 has been modified accordingly.

“Lines 534-535, include the Figure reference”.

Figure reference has been included (line 612 of the current version).

Reviewer 2 Report

In this manuscript, Ecker et al report methods to express influenza virus HA and NA proteins in human cells and the characterisation of the expressed proteins. Expression of recombinant HA and NA is not exactly new, even if the focus so far has often been on insect cell systems. Despite this, a more detailed description is useful for the influenza virus research and influenza vaccine fields, as previous work has frequently glanced over the methodological detail in a rather cavalier fashion.

I recommend that the authors address the points below to improve the usefulness of their manuscript and correct a few errors.

Major point:

  1. The main point I want to raise is surprising, giving the authors’ attention to detail throughout large parts of the manuscript and their obvious aim to provide a useful guide to production of recombinant influenza virus glycoproteins: The results section lacks a comparison between expression systems using transient transfection and stably transfected cell lines. In fact, it is not even clear which expression system was used for the generation of the proteins shown in the characterisation data (Figures 4 – 10, except Figure 4b). Please provide comparative data on protein yields for the two systems and describe which expression system was used for each experiment/protein characterisation reported.

Minor points:

  1. Line 47: ‘benefits for’ should be ‘benefits of’
  2. Lines 55 – 56: incomplete sentence.
  3. Lines 94ff: Please describe in more detail the exact truncation points of the HA and NA genes so that other researchers could recreate the gene constructs. Perhaps a supplementary figure with aligned sequences might help.
  4. Legend to Figure 1: The colour descriptions do not match the colours I see in a printed copy of the manuscript, nor the colours displayed on my monitor. Please double-check and make unequivocal (can colour patches or coloured letters be inserted into the legend?).
  5. Lines 298 – 299: The description of the determination of the HA titre appears to be wrong: Shouldn’t the titre be the reciprocal of the last dilution that shows complete agglutination?
  6. Line 379: The authors talk about ‘subtype-specific … mAbs’, but according to other parts of the manuscript, not least the legend to Figure 5, these mAbs are ‘clade’-specific, which may be broader than subtype-specific. I am not 100% sure what the authors mean by clade-specific, but assume they mean specific for the two phylogenetic groups of influenza A virus HAs. Please clarify what exactly is meant and use consistent terminology.
  7. Line 434: ‘…cold-chain may not feasible’: the word ‘be’ is missing between ‘not’ and ‘feasible’.
  8. Legend to Figure 7, lines 450 – 451: Something seems to be missing, the sentence is grammatically incomplete and the meaning is not clear.
  9. Lines 458 – 459: There seems to be a misunderstanding of the NA activity assay (which is essentially the ELLA, to which a reference should be added): peanut agglutinin does not bind cleaved sialic acid (which would be washed away!), but remaining Gal residues on the fetuin.
  10. Lines 459ff: It is hard to make a comparison between NA activities displayed by recombinant NA and NA on virions, without quantitation of the latter at the protein level. Thus, the authors should be cautious in their statement (‘the magnitude of NA activity featured by NA recombinant proteins is similar to those exerted by the corresponding NA on the influenza viral particles’); what is possible to say is that the order/ranking of activity levels is similar between NA on virions and recombinant NA.
  11. Legend to Figure 9, line 488: What are ‘three-fold serial dilutions concentrations’? I think ‘concentrations’ should be deleted.
  12. Figure 10: Panel B seems to be cut off and its description in the legend is unclear. I don’t really know what I am looking at in panel B. Where is the H3N2 staining? Also, does ‘D’ in the legend refer to panel C? And ‘C’ (line 518) to panel B?
  13. Discussion, lines 547ff: Did the authors consider, or even test, conformational changes induced by acid treatment of HA? Would this have been a useful addition to the heat treatment?
  14. Line 549: ‘relative rHA’ should be ‘relative to rHA’, I believe.
  15. Line 583: ‘though’ should be ‘through’.
  16. Line 586: ‘polyclonal and mAbs’ – something is missing here; ‘polyclonal antibodies and mAbs’ would read better.
  17. Please use the commonly used conventions for naming influenza virus strains: years before 2000 are shown as the last two digits only (as in ‘A/Chile/1/83’), years from 2000 onwards are shown as the full year (as in ‘A/Brisbane/59/2007’).
  18. There are a number of singular/plural errors, such as in lines 115 (‘plasmids was’), 358 (‘trimerization and a tetramerization domains’), 484 (‘a murine mAbs’), 489 (‘rHAΔSA protein enable’), and more.

Author Response

Dear Reviewer,

First of all, we would like to thank you for your valuable comments that contributed to improve the overall quality of our manuscript.

Please below you can find a point by point response (highlighted in bold) to your comments. We hope you will find them satisfying.

Major point:

“The main point I want to raise is surprising, giving the authors’ attention to detail throughout large parts of the manuscript and their obvious aim to provide a useful guide to production of recombinant influenza virus glycoproteins: The results section lacks a comparison between expression systems using transient transfection and stably transfected cell lines. In fact, it is not even clear which expression system was used for the generation of the proteins shown in the characterisation data (Figures 4 – 10, except Figure 4b). Please provide comparative data on protein yields for the two systems and describe which expression system was used for each experiment/protein characterisation reported”.

All the figures have been generated from data obtained by proteins/cells derived from stably-transfected cell lines. We aimed at describing and sharing with the scientific community a methodology to produce influenza virus recombinant proteins by the use of stably-transfected cell lines as a principal mean of reducing the associated costs with transient transfection. We thank the reviewer for having raised this concern and thus giving us the opportunity to make it clearer. The figure legends of all the figures have been modified accordingly.

Minor points:

We thank the reviewer for having noticed all the below listed typos and grammatical errors and for each amendment we reported the corresponding lines on the current version of the manuscript.

“Line 47: ‘benefits for’ should be ‘benefits of’”.

Amended (line 49).

“Lines 55 – 56: incomplete sentence”.

Amended (line 62).

“Lines 94ff: Please describe in more detail the exact truncation points of the HA and NA genes so that other researchers could recreate the gene constructs. Perhaps a supplementary figure with aligned sequences might help”.

The exact truncation points have been added (lines 120-123; 129-131).

“Legend to Figure 1: The colour descriptions do not match the colours I see in a printed copy of the manuscript, nor the colours displayed on my monitor. Please double-check and make unequivocal (can colour patches or coloured letters be inserted into the legend?)”.

We thank the reviewer for having noticed that. Figure 1 has been modified accordingly.

“Lines 298 – 299: The description of the determination of the HA titre appears to be wrong: Shouldn’t the titre be the reciprocal of the last dilution that shows complete agglutination?”

We thank the reviewer for having noticed that. The sentence has been modified accordingly (line 381).

“Line 379: The authors talk about ‘subtype-specific … mAbs’, but according to other parts of the manuscript, not least the legend to Figure 5, these mAbs are ‘clade’-specific, which may be broader than subtype-specific. I am not 100% sure what the authors mean by clade-specific, but assume they mean specific for the two phylogenetic groups of influenza A virus HAs. Please clarify what exactly is meant and use consistent terminology”.

We thank the reviewer for having noticed that. The text in the main manuscript and the legend to figure 5 have been modified accordingly (lines 475-476; 503-504).

“Line 434: ‘…cold-chain may not feasible’: the word ‘be’ is missing between ‘not’ and ‘feasible’”.

Amended (line 541).

“Legend to Figure 7, lines 450 – 451: Something seems to be missing, the sentence is grammatically incomplete and the meaning is not clear”.

We thank the reviewer for having noticed that. Legend to figure 7 has been modified accordingly (lines 556-559).

“Lines 458 – 459: There seems to be a misunderstanding of the NA activity assay (which is essentially the ELLA, to which a reference should be added): peanut agglutinin does not bind cleaved sialic acid (which would be washed away!), but remaining Gal residues on the fetuin”.

We thank the reviewer for having noticed that. The sentence has been modified accordingly (line 567) and reference for ELLA has been added in the M&M section (lines 385-386).

“Lines 459ff: It is hard to make a comparison between NA activities displayed by recombinant NA and NA on virions, without quantitation of the latter at the protein level. Thus, the authors should be cautious in their statement (‘the magnitude of NA activity featured by NA recombinant proteins is similar to those exerted by the corresponding NA on the influenza viral particles’); what is possible to say is that the order/ranking of activity levels is similar between NA on virions and recombinant NA”.

The sentence has been modified accordingly (line 567).

“Legend to Figure 9, line 488: What are ‘three-fold serial dilutions concentrations’? I think ‘concentrations’ should be deleted”.

The legend to figure 9 has been modified accordingly (line 604).

“Figure 10: Panel B seems to be cut off and its description in the legend is unclear. I don’t really know what I am looking at in panel B. Where is the H3N2 staining? Also, does ‘D’ in the legend refer to panel C? And ‘C’ (line 518) to panel B?”

We thank the reviewer for having noticed that. Figure 10 and its legend have been modified accordingly.

“Discussion, lines 547ff: Did the authors consider, or even test, conformational changes induced by acid treatment of HA? Would this have been a useful addition to the heat treatment?”

pH-induced conformational changes of rHA produced though the methodology described in this manuscript have been previously described in our recent published manuscript (Sautto et al. 2020. J. Immunol.). We appreciate the reviewer’s interest in that and we now have included this point in the discussion section (lines 682-684).

“Line 549: ‘relative rHA’ should be ‘relative to rHA’, I believe”.

Amended (line 675).

“Line 583: ‘though’ should be ‘through’”.

Amended (line 713).

“Line 586: ‘polyclonal and mAbs’ – something is missing here; ‘polyclonal antibodies and mAbs’ would read better”.

Amended (line 716).

“Please use the commonly used conventions for naming influenza virus strains: years before 2000 are shown as the last two digits only (as in ‘A/Chile/1/83’), years from 2000 onwards are shown as the full year (as in ‘A/Brisbane/59/2007’)”.

Influenza strain nomenclature has been modified in the whole manuscript accordingly.

“There are a number of singular/plural errors, such as in lines 115 (‘plasmids was’), 358 (‘trimerization and a tetramerization domains’), 484 (‘a murine mAbs’), 489 (‘rHAΔSA protein enable’), and more”.

Errors have been corrected and the manuscript has been entirely revised for possible additional grammatical errors accordingly.

Reviewer 3 Report

The authors present a detailed method for producing a recombinant influenza vaccine.

In order to make a product into a viable vaccine alternative, the product has to go through a scrutinizing process, from efficacy, safety, and compliance, to product cost, yield, stability consistency, scalability, and lastly regulatory issues. Some of these topics have been covered in the manuscript.

A recent review in this journal is very relevant to the submitted manuscript;

Prospects and Challenges in the Development of Universal Influenza Vaccines. Madsen A, Cox RJ. Vaccines (Basel). 2020 Jul 6;8(3):E361. doi: 10.3390/vaccines8030361. PMID: 32640619

Maybe the authors could outline how the described method could be used to incorporate other influenza antigens (e.g., M, NP) to broaden the immune response and for a potential universal vaccine?

How do the authors see the recombinant vaccine replace the conventional egg grown vaccine in the future?

Author Response

Dear Reviewer,

First of all, we would like to thank you for your valuable comments that contributed to improve the overall quality of our manuscript.

Please below you can find a point by point response (highlighted in bold) to your comments. We hope you will find them satisfying.

“The authors present a detailed method for producing a recombinant influenza vaccine.

In order to make a product into a viable vaccine alternative, the product has to go through a scrutinizing process, from efficacy, safety, and compliance, to product cost, yield, stability consistency, scalability, and lastly regulatory issues. Some of these topics have been covered in the manuscript.

A recent review in this journal is very relevant to the submitted manuscript;

Prospects and Challenges in the Development of Universal Influenza Vaccines. Madsen A, Cox RJ. Vaccines (Basel). 2020 Jul 6;8(3):E361. doi: 10.3390/vaccines8030361. PMID: 32640619.”

We thank the reviewer for having suggested this very informative and recent review article which we read with great interest and considered for additional discussion points in the revised version of our manuscript.

“Maybe the authors could outline how the described method could be used to incorporate other influenza antigens (e.g., M, NP) to broaden the immune response and for a potential universal vaccine?”

We included this points in the discussion (lines 721-726).

“How do the authors see the recombinant vaccine replace the conventional egg grown vaccine in the future?”

We included this points in the introduction and in the discussion (lines 52-59; 721-730).